# Fertilization for Growth or Feeding the Weeds? A Deep Dive into Nitrogen’s Role in Rice Dynamics in Ecuador

**DOI:** 10.3390/life14121601

**Published:** 2024-12-04

**Authors:** Cristhian Fernando Sánchez-Sabando, Adriana Beatriz Sánchez-Urdaneta, Fernando David Sánchez-Mora, Gary Eduardo Loor-Escobar, Barlin O. Olivares

**Affiliations:** 1Postgraduate Studies Faculty, Technical University of Manabí, Portoviejo 130105, Ecuador; cristhian.sanchez@upl-ltd.com; 2Sustainable Agriculture and Bioenergy Group, Research Division, Faculty of Agricultural Engineering and Health Sciences, Technical University of Manabí, Portoviejo 130105, Ecuador; adriana.sanchez@utm.edu.ec; 3Department of Botany, Faculty of Agronomy, University of Zulia, Maracaibo 4005, Venezuela; 4Sustainable Agriculture and Bioenergy Group, Faculty of Agricultural Engineering, Technical University of Manabí, Portoviejo 130105, Ecuador; fernando.sanchez@utm.edu.ec; 5Research Division, Technical University of Manabí, Portoviejo 130105, Ecuador; eduardo.loor@utm.edu.ec; 6Biodiversity Management Research Group (GESBIO-UCO), Rabanales Campus, University of Córdoba, National Highway IV km 396, 14014 Córdoba, Spain

**Keywords:** abiotic stresses, agronomic traits, nitrogen fertilization, rice yield, weed dynamics

## Abstract

Rice (*Oryza sativa* L.) is a crucial crop for employment and agricultural output and heavily reliant on family labor. This study evaluated the effects of nitrogen levels (80, 120, and 160 kg·ha^−1^) on weed incidence and key agronomic variables, including vegetative growth, yield, and related traits, in Ecuador’s primary rice-growing regions, Guayas and Los Ríos. A split-plot randomized complete block design was implemented using two rice varieties (INIAP-FL-Elite and SFL-11) and three planting densities (20 × 30, 25 × 30, and 30 × 30 cm). Weed incidence was higher in Los Ríos, dominated by grasses (55.28%), while Cyperaceae (46.27%) prevailed in Guayas. The data analysis included non-parametric tests to identify significant treatment effects, debiased sparse partial correlations (DSPCs) to reveal key agronomic interactions, and principal component analysis (PCA) to identify influential traits, ensuring robust and normalized interpretations. Analysis of variance indicated significant effects for all variables, with vegetative growth (VG) most affected (*p* < 0.001). The number of panicles (NP) and days to flowering (DF) showed significant though less pronounced effects, while the panicle length (LP) and 1000-seed weight (TSB) exhibited moderate responses. The DSPCs highlighted the grains per panicle (GP) and total biomass (SB) as critical variables, with significant correlations between the days to flowering and the tiller count at 55 days (r = 0.750, *p* < 0.001) and between the vegetative growth and the total biomass (r = 0.678, *p* < 0.001). PCA explained 58.8% of the total variance, emphasizing the days to flowering, plant height, total biomass, and yield as the most influential traits. These findings underline the importance of integrated nutrient and weed management strategies tailored to Ecuador’s agroecological conditions.

## 1. Introduction

In Ecuador, there is a considerable rice germplasm comprising publicly released rice varieties, nursery observation lines, and landraces varieties [1]. A total of 317,400 ha of rice (*Oryza sativa* L.) are cultivated and distributed across the provinces of Guayas, Los Ríos, Manabí, Loja, and El Oro, with a total production of 1,697,353 tons, of which Los Ríos and Guayas provide 95% of the national output [2]. This crop is a significant source of employment, with 89,896 people working in primary production, mostly using family labor. Additionally, in 2021, rice cultivation contributed 39% of the gross agricultural added value [3].

Conventional and organic rice farming systems differ significantly in their environmental impacts, particularly in the context of climate change. Traditional agriculture, heavily dependent on chemical inputs, such as synthetic fertilizers and pesticides, is a major contributor to greenhouse gas (GHG) emissions, including carbon dioxide (CO_2_), methane (CH_4_), and nitrous oxide (N_2_O); these emissions are notably exacerbated in flood-irrigated systems, which promote CH_4_ production [4,5]. The same authors mentioned that, in contrast, organic farming practices reduce CO_2_ emissions by approximately 32.7% compared to conventional methods using natural fertilizers, such as vermicompost and livestock manure. However, during the initial transition to organic systems, higher emissions of N_2_O and CH_4_ may occur due to the decomposition of organic matter [4,5].

From a sustainability perspective, organic practices offer several environmental benefits. They lower environmental toxicity, mitigate freshwater eutrophication, reduce fossil fuel dependency, and improve soil properties and microbial diversity. These practices also enhance nutrient use efficiency while minimizing emissions of ammonia (NH_3_), sulfur dioxide (SO_2_), and nitrogen dioxide (NO_2_) [6]. Although organic systems require time to achieve stabilization, they represent a viable alternative for mitigating climate change and fostering a resilient global agricultural framework [5].

Rice production, like other crops, depends on the management practices employed by farmers, especially the correct use of herbicides. The climate and soil conditions in Ecuador’s coastal region provide favorable agro-edaphoclimatic conditions for the development of weeds, such as *Cyperus rotundus*, *Panicum maximum*, and *Rottoboellia exaltata*, which aggressively grow and pose challenges for effective management [7]. Various studies have shown that long-term fertilization significantly affects the density and composition of weed communities, thus influencing the productivity of rice crops [8,9,10,11]. On the other hand, mechanical weed management has demonstrated notable benefits in rice growth by increasing the tiller number, SPAD values, and total biomass, leading to higher grain yields. Moreover, appropriate plant spacing in rice has been identified as a crucial factor in mitigating competition with weeds and optimizing crop yield [12,13,14,15,16].

Plant spacing in rice cultivation significantly impacts weed community density and composition. Recent studies have shown that narrower spacing between rice plants can reduce weed infestation by limiting the resources available for their growth [17]. However, weeds’ response to plant spacing may vary depending on the local conditions and the species of weeds present, requiring site-specific evaluations [18,19].

In Ecuador, in rice fields close to the Guayas River basin, common herbicides, such as butachlor and pendimethalin, used to control grassy weeds [20], are applied pre-emergence or post-emergence in flooded rice fields or directly seeded. According to Nazir et al. [21], several rice systems are used, such as transplanting, direct seeding, and systems of intensification, of which penoxsulam herbicide recorded the highest grain yield and was effective in reducing weed populations and its dry biomass per square meter.

Balanced fertilization plays an important role in integrated weed management in rice cultivation. The rational application of nutrients, especially nitrogen, phosphorus, and potassium, promotes vigorous crop growth, increasing its competitive ability against weeds [22]. Furthermore, different fertilization treatments can influence weed community composition, with some species responding differentially to the availability of specific nutrients [23].

The integrated management of plant spacing and fertilization offers an effective strategy for weed control in rice cultivation. Combining optimal plant spacing and balanced fertilization improves crop yield and reduces weed density and biomass [24]. However, it is important to consider that these practices must be adapted to the specific conditions of each cropping system and region, as the effects may vary depending on the environmental factors and weed species present [25].

However, the current situation in Ecuador presents a challenge: the intensification of nitrogen fertilization, a common practice to boost rice growth, may also inadvertently feed the weeds. Studies have shown that long-term fertilization alters weed community composition and density, with certain species thriving in nitrogen-rich environments. This poses a dual threat—reduced crop productivity and increased costs associated with herbicide use [8,26]. Furthermore, mechanical and cultural practices, such as adjusting plant spacing, have shown promise in mitigating weed competition, but their effectiveness varies depending on the local conditions and the weed species present [27,28]. If not managed properly, this imbalance can lead to decreased rice yields and environmental degradation due to increased chemical inputs [29,30].

The consequences of ineffective weed control in nitrogen-rich environments are clear: reduced rice yields, increased costs for farmers, and potential long-term soil degradation [31,32]. Given that rice is essential to both Ecuador’s food security and its rural economy, failing to address these challenges could have widespread socioeconomic and environmental impacts. As nitrogen fertilization is a key component of rice cultivation, it is critical to understand its role in weed dynamics to develop more sustainable management practices.

While prior research has explored nitrogen’s role in rice productivity, it often overlooks its nuanced effects on specific weed species and the interaction with key agronomic variables under distinct regional conditions. This study addresses these gaps by investigating how nitrogen fertilization, at various levels, modulates weed dynamics and rice performance in Ecuador’s unique agroecosystems. By applying advanced analytical methods like debiased sparse partial correlations (DSPCs), principal component analysis (PCA), and the Kruskal–Wallis test, we aim to clarify these complex interactions, enhancing our understanding of nitrogen’s dual impact on both crop productivity and weed dynamics.

Therefore, this study seeks to explore the interactions among nitrogen fertilization, plant spacing, and weed management in rice cultivation in Ecuador. Specifically, we ask “Does nitrogen fertilization in rice fields promote crop growth, or does it inadvertently feed the weeds, thereby reducing yield potential?” By addressing this question, we aim (i) to identify whether nitrogen levels influence the incidence of weeds and (ii) to identify whether nitrogen fertilization influences the main variables: vegetative growth, yield, and other characteristics of rice plants.

It is hypothesized that nitrogen fertilization in rice fields enhances crop growth and, when combined with appropriate planting distances and effective weed management strategies, it can mitigate the competitive impact of weeds on yield by reducing their incidence.

## 2. Materials and Methods

### 2.1. Location and Study Period

This research was conducted in two rice-growing areas of Ecuador during the dry season of 2022 (August to December). The locations were: Virgen de Fátima Parish, Yaguachi Municipality, Guayas Province, located in the South Coast Experimental Station (INIAP), between the coordinates 2°15′15″ S and 79°30′40″ W, and in the Montalvo Parish, Babahoyo Municipality, Los Ríos Province, located in the CEDEGE area, at the coordinates 1°53′31.38″ S and 79°27′27.31″ W.

### 2.2. Treatments in Study

The factors evaluated were the localities (Virgen de Fatina and Montalvo), nitrogen doses at three levels (80, 120, and 160 kg·ha^−1^), rice varieties (*O. sativa*) at two levels (INIAP-FL-Elite and SFL-11), and planting distances at three levels (20 × 30, 25 × 30, and 30 × 30 cm). These combinations of nitrogen doses, rice varieties, and planting distances formed the treatment groups.

Before the application of treatments, weeds in the study plots were classified based on their leaf width up to the family level, as reproductive structures were absent at the time of evaluation. Narrow-leaved weeds were identified and categorized under the botanical families Cyperaceae and Gramineae. Furthermore, broad-leaved weeds were grouped and classified based on their structural characteristics as herbaceous, semi-woody, or woody.

### 2.3. Experimental Design

A total of 108 plots were used for the study in each locality, with each plot covering a total area of 9.00 m^2^. The width of the plots was 1.80 m, and the length was 5.00 m. The usable area for each plot was 6.00 m^2^, with a separation of 1.5 m between blocks, corresponding to a total experimental area of 162 m^2^.

A randomized complete block design with split-plot treatments was employed, with the main plot representing the location, and a full factorial arrangement (2 × 3 × 2 × 3) applied in the subplots with three replications. The main plot was composed of the location (two levels), while the factors and levels within the subplots included nitrogen, rice varieties, and planting distances as previously mentioned.

### 2.4. Management of the Trial

Before land preparation, soil sampling was performed at both the experimental site for physical and chemical analysis. A zigzag sampling method was used, with samples taken from several points in the field to create a composite sample, ensuring representativeness. Analyses were conducted at the National Institute for Agricultural Research, El Boliche Experimental Station. In both sites, the soil is clay loam.

To establish the seedbeds, 3 kg of certified seed was used for each variety under study. Seedbeds were prepared with an area of 10 m^2^, and 300 g of seed per m^2^ were distributed in each bed. Vydate (2 cc·m^−2^) was applied to prevent nematode damage. Fertilization was performed on the seedbeds 12 days after sowing (DAS), with urea applied at a rate of 20 g·m^−2^. Before transplanting, the seedbeds were exposed to a water layer of approximately 5 to 8 cm for 4 days to facilitate seedling removal and reduce root damage. Transplanting was carried out when the plants were 25 days old in flooded soil without water lamina to prevent damage from the apple snail (*Pomacea canaliculata*). Between 3 and 4 plants per planting site were transplanted, considering the planting distances in this study (20 × 30, 25 × 30, and 30 × 30 cm).

Weed control was performed 10 days after transplanting (DAT) when the weeds had developed 2 to 3 leaves. A chemical control method was employed, using selective herbicides for rice cultivation, applying butachlor + pendimethalin + bispyribac at rates of 40 + 25 + 01 L commercial product·ha^−1^, respectively. Then, weed monitoring was conducted at 30, 45, and 60 days after herbicide application (DAA). Fertilization was applied at 15, 30, and 45 days after transplanting (DAT), with nitrogen sources provided according to the nitrogen levels specified in the treatments. Harvesting occurred 120 days into the crop cycle when the materials reached physiological maturity.

### 2.5. Evaluation of Study Variables

Thirteen agronomic variables were evaluated to examine the vegetative growth, yield performance, and other characteristics of the rice plants (Table 1). The vegetative cycle was expressed in days (VG), considering the time lapse from seedling emergence to the harvest of all experimental units. The variables evaluated included the number of days to flowering (DF), counted from transplanting to the appearance of the first panicles. Plant height (PH) was measured before harvest in 10 randomly selected plants from the usable area of each experimental unit, measured from the plant collar to the apex of the most prominent panicle, with the data expressed in cm. The number of tillers was counted 55 days after transplanting (NT55D) and at harvest (NHT) in the 10 selected plants. The number of panicles (NP) was recorded, and panicle length (LP) was measured from the ciliary node to the panicle apex, excluding awns. Measurements were made on panicles from the 10 selected plants. The grain sterility percentage was determined on the panicles selected for the panicle length variable, and the total number of fertile (filled) (GP) and sterile (empty) grains (KS) was counted. The results are expressed as percentages. Finally, 1000 grains from each treatment were counted and weighed (TSB) using an analytical balance, with the results expressed in grams.

Each plot was harvested when the plants completed a productive cycle, and the weight (WS) and moisture content (MH) of the samples were measured. Subsequently, the yield (YHA) was expressed in kilograms per hectare, estimated although the formula WAT = [(WS × (100 − IM)/(100 − DM)) × (10/HA)], where WS = the weight of the sample (g), IM = the initial moisture content; DM = the desired moisture content of 14%; HA = the harvested area (m^2^); WAT = the weight adjusted to the treatment.

### 2.6. Data Analysis

To improve normality and reduce the influence of outliers, a log transformation (base 10) was applied to the data in version 4.0.2 of R software (R Core Team, Vienna, Austria) [33] using the “glog” function [34]. This transformation ensured that variables with skewed distributions were normalized, allowing for more robust statistical analyses and a better fit for parametric models.

Auto-scaling was performed to standardize the variables and eliminate disparities in scale among different units. This procedure involved mean-centering the data and dividing by the standard deviation of each variable. Auto-scaling ensured that all variables contributed equally to multivariate analyses and helped mitigate potential biases due to differences in measurement units or magnitude.

To evaluate the significance of the effects of nitrogen fertilization and rice variety on the measured agronomic variables, a non-parametric approach was utilized due to deviations from normality in certain variables. Specifically, the Kruskal–Wallis test, a non-parametric version of ANOVA, was employed to assess the differences across the treatment groups without assuming a normal distribution of the data [35]. This test ranked the data and compared the distribution of ranks across the different nitrogen fertilization levels and rice varieties. The Kruskal–Wallis Test was selected to provide a more robust analysis under the assumption of non-normality, ensuring that significant differences among groups were accurately captured. Additionally, false discovery rate (FDR) correction was applied to account for multiple comparisons and reduce the risk of type I errors. The test was performed in R using the “Kruskal Test” function in R.

In this study, the debiased sparse partial correlations (DSPC) network was applied to identify the complex interactions among variables such as vegetative growth, yield characteristics, and nitrogen levels, providing insights into the underlying agronomic dynamics. This analysis was conducted using the *huge* package in R, which is designed to estimate high-dimensional sparse graphs and infer partial correlations among the observed variables, resulting in a more robust network structure that highlights the most relevant agronomic factors [36].

Principal component analysis (PCA) was applied to reduce the dimensionality of the dataset and identify the most influential agronomic variables in explaining the variance across different nitrogen fertilization levels and rice varieties. Variables with loading values greater than 0.30 were considered significant contributors to the principal components, following common criteria in multivariate analysis [37,38]. This threshold allowed us to identify key variables driving the variability in the data. In R, the “prcomp” function was used to perform PCA, with the “scale = True” argument to ensure proper standardization. The “summary” function provided the proportion of variance explained by each principal component, and the “biplot” visualized the components and loadings.

## 3. Results

### 3.1. Levels of Weed Incidence

Table 2 presents the weed incidence data from two locations: Virgen de Fatima, Guayas and Montalvo, Los Rios. Before applying the treatments, the weeds in the study plots were categorized based on leaf width. Two distinct groups were identified: narrow-leaved weeds, represented by botanical families such as Cyperaceae and Gramineae, and broad-leaved weeds, which were further classified according to their herbaceous, semi-woody, or woody structure (Table 2).

In both localities, three main categories of weeds were observed: grasses, sedges, and broadleaves. However, notable differences in their incidence were detected between the two areas. Statistically significant differences (*p* < 0.001) were found among these groups in Virgen de Fatima, Guayas, where sedges constituted 46.27% of the species, grasses 39.22%, and broadleaves 14.61%. In contrast, in Montalvo, Los Ríos, grasses dominated (55.28%), followed by sedges (34.56%) and broadleaves (10.22%), with similarly significant statistical differences (*p* < 0.001) among the three groups (Table 2).

In both locations, broadleaves had the lowest incidence. In Virgen de Fatima, Guayas, Cyperaceae and grasses had 3.07 and 2.60 times more incidence than broadleaves, respectively, while Cyperaceae were 1.18 times more abundant than grasses. In Montalvo, Los Ríos, grasses and sedges had 5.62 and 3.50 times more incidence than broadleaves, with grasses exceeding sedges by 1.57 times.

No herbicidal toxicity was observed during this study. However, in Virgen de Fatima, Guayas, with the SFL-11 variety under the application of 160 kg·ha^−1^ of N and a planting density of 30 × 30 cm, 25% weed presence was recorded both at 25 days post-planting and at harvest. This weed population predominantly consisted of species within the Gramineae and Cyperaceae families (Appendix A, Table A1).

### 3.2. Effects of Nitrogen Levels on Rice Plants

#### 3.2.1. Non-Parametric ANOVA

The results of the Kruskal–Wallis test demonstrate significant differences among the treatments for the measured agronomic variables (Figure 1a). Vegetative growth (VG) exhibited the highest F-statistic (33.079), indicating a strong effect of the treatments on this variable, with an extremely low *p*-value (6.56 × 10^−8^), confirming its high statistical significance even after FDR adjustment (8.53 × 10^−7^) (Figure 1b). Other variables, such as the number of panicles (NP) (Figure 1c) and days to flowering (DF) (Figure 1d), also showed significant differences, with F-statistics of 12.982 and 10.883, respectively. Their *p*-values were both below the 0.01 threshold, confirming the significant effects of nitrogen fertilization and variety on these parameters.

In contrast, the panicle length (LP) (Figure 1e) and 1000 seed weight (TSB) (Figure 1f) showed moderately significant effects, with F-statistics of 8.3256 and 8.2106, respectively. Both had *p*-values slightly above 0.01 but remained statistically significant after FDR correction, with values of 0.042 for both variables. These findings suggest that while vegetative growth was most strongly impacted by the treatments, factors like panicle length and seed biomass were also influenced, though to a lesser extent.

#### 3.2.2. Debiased Sparse Partial Correlation (DSPC) Network

The results of the DSPC network analysis identified key agronomic variables with central roles in the network based on their degree and betweenness centrality measures (Figure 1). The grains per panicle (GP) and moisture at harvest (MH) demonstrated the highest degree values (five connections each), indicating that these variables are highly interconnected with other traits in the network. Notably, GP had the highest betweenness value (20), suggesting that it serves as a critical hub in the network, facilitating interactions among other agronomic factors. Similarly, the sample biomass (SB) and vegetative growth (VG) also exhibited moderate centrality, with four connections each, indicating their importance in the overall structure of the agronomic system under study.

In terms of the partial correlations, several significant relationships were observed, particularly those with coefficients greater than 0.45. The strongest positive correlation was identified between the days to flowering (DF) and several tillers at 55 days (NT55D) (0.750, *p* = 9.17 × 10^−6^), suggesting a direct relationship between these two growth stages. Additionally, vegetative growth (VG) showed moderate positive correlations with plant height (PH) (0.457, *p* = 0.00184) and sample biomass (SB) (0.678, *p* = 2.76 × 10^−5^), indicating that vegetative development significantly impacted both plant size and overall biomass production. The panicle length (LP) was also positively associated with the moisture at harvest (MH) (0.517, *p* = 1.43 × 10^−5^), highlighting the potential influence of panicle size on moisture retention at the end of the growing season.

Furthermore, the number of tillers at 55 days (NT55D) had a strong positive relationship with the grains per panicle (GP) (0.666, *p* = 1.28 × 10^−4^), underscoring the role of early tiller development in determining grain yield potential. The significant *p*-values associated with these partial correlations (*p* < 0.05) provide robust statistical support for these interactions, reinforcing the importance of the GP, MH, and VG as central variables in the network. These findings suggest that these key agronomic traits are likely influenced by nitrogen fertilization levels and could serve as critical targets for optimizing rice growth and yield in future studies (Figure 2).

#### 3.2.3. Principal Component Analysis (PCA)

The pairwise score plots of the selected principal components (PC1, PC2, and PC3) are presented in Figure 3, contributing to a total explained variance of 58.8% (Figure 3a). The distribution of the data points across the principal components indicates some degree of separation, particularly along PC1, which captures the largest proportion of variability. However, there is substantial overlap among the nitrogen treatments, suggesting that the differentiation among the nitrogen doses is not fully explained by these three principal components alone.

The plot shows some separation among the nitrogen treatments, though there is considerable overlap, particularly between the green (160 kg·ha^−1^) and blue (80 kg·ha^−1^) groups, suggesting that the two nitrogen doses exhibited similar responses along these two principal components (Figure 3b). The red group (120 kg·ha^−1^) shows more spread, with a few outliers (e.g., R19), indicating more variability in its response.

The loading values for PC1 (Figure 3c) indicate that the days to flowering (DF), plant height (PH), kernel spacing (KS), sample biomass (SB), and yield per hectare (YHA) were the most influential variables along this component. The negative loadings for these variables suggest that higher values of PC1 were associated with lower values for these agronomic traits. In particular, the DF and PH exhibited the strongest negative influence, meaning that as PC1 increased, flowering occurred later and the plants were shorter. Additionally, the negative influence of YHA indicates that higher yields were associated with lower values of PC1, suggesting a potential trade-off between vegetative growth and yield performance, influenced by nitrogen levels.

In PC2, the variables of the number of tillers at 55 days (NT55D), number of panicles (NP), and grains per panicle (GP) were the most significant contributors, with GP showing the strongest negative correlation. The positive loadings for NT55D and NP indicate that higher values of PC2 were associated with increased tillering and panicle number, while the negative loading of GP suggests a decrease in the grain count per panicle as PC2 increased. This result highlights how early tillering and panicle formation are critical traits that drive variability along PC2, while grain production per panicle may be compromised under certain conditions, particularly as nitrogen treatments impact tillering dynamics.

PC1, which explains 29.4% of the variance (Figure 3d), was primarily driven by traits such as the days to flowering (DF), plant height (PH), sample biomass (SB), and yield per hectare (YHA), which are positioned along the negative side of the PC1 axis. This indicates that these variables negatively influenced PC1, meaning that higher values of PC1 were associated with lower values for these traits. In contrast, the panicle length (LP) and kernel spacing (KS) made positive contributions to PC1, suggesting a positive association with higher values of PC1.

On the other hand, PC2, which accounted for 15.2% of the variance (Figure 3d), was heavily influenced by the number of tillers at 55 days (NT55D), the number of panicles (NP), and the grains per panicle (GP). NT55D and NP were positively associated with PC2, while GP showed a negative association, indicating an inverse relationship with these traits. The positioning of the points representing the observations (colored based on the nitrogen treatments) shows a degree of overlap, but certain patterns suggest that PC1 and PC2 captured some of the variance related to the nitrogen dose effects on these key agronomic variables.

## 4. Discussion

These findings reflect the complex interactions between nitrogen fertilization and key agronomic traits, influencing both vegetative growth and reproductive outcomes.

### 4.1. Levels of Weed Incidence

Lastly, there were notable differences between the two locations. Los Rios showed a generally higher incidence of Gramineae weeds compared to Las Guayas, possibly due to environmental conditions that favor grass weed proliferation. However, Cyperaceae incidence was similar in both locations, indicating that these weeds are less influenced by geographic factors. Broadleaf weeds were slightly more prevalent in Las Guayas than in Los Rios, suggesting that broadleaf species might respond to location-specific factors like soil or moisture conditions.

The variation in the weed composition between the two localities can be attributed to differences in the agroecological conditions and management practices employed in the study areas. Factors such as soil type, temperature, water availability, and crop rotation further influenced the population dynamics of the weeds [39].

In rice cultivation, as in many other crops, weeds cause significant damage by competing for nutrients, water, light, and space. This competition weakens the crop, making it more vulnerable to pests and diseases throughout all growth stages. It also reduces yields, contaminates seeds, and, in severe cases, leads to crop failure [40].

In both localities (Guayas and Los Ríos), grasses and sedges consistently dominate over broadleaf weeds, suggesting that these weed types are better adapted to rice field conditions. The differences between the two provinces could be attributed to variations in agricultural management practices, soil type, and agroclimatic factors, which influence the population dynamics of the weeds as well as the plasticity of their root systems, allowing them to cope with flood–dry cycles. The treatments applied do not appear to significantly alter the dominance of grasses and sedges over broadleaves [41,42].

Chemical control through the use of selective herbicides for rice offers significant advantages over other weed control methods, as it can eliminate weeds either before or after emergence, provides broad-spectrum control, and has lasting residual effects [43]. The sequential application of pre- and post-emergence herbicides, such as pendimethalin followed by bispyribac-sodium, has been shown to enhance weed control, improve weed management efficiency, and increase grain yield in aerobic rice systems [44].

The classification of weeds in the experimental plots based on leaf width allowed for a clear distinction between narrow-leaved species, such as those from the Cyperaceae and Gramineae families, and broad-leaved species, which were further categorized based on their herbaceous, semi-woody, and woody structures. In Guayas province, Cyperaceae species dominated, accounting for 46%, followed by Gramineae at 39%, and broadleaf species at 15%. These differences in species prevalence suggest that Cyperaceae, in particular, possess a higher competitive ability in the local environment, possibly due to their greater tolerance for resource competition, such as water and nutrients. These findings align with previous studies indicating that agronomic practices and soil management can influence the composition of plant communities [45,46,47].

In contrast, in Los Ríos province, grasses were more dominant, representing 55% of the species, followed by sedges (35%) and broadleaves (10%). The statistical differences observed between the two provinces reveal a differentiated species prevalence, likely influenced by variations in agricultural practices, fertilization regimes, and soil characteristics, all of which impact the growth and competitiveness of weed species. These results echo studies’ findings that demonstrate how fertilization and pest control affect weed community composition [9,48,49,50]. Notably, nitrogen fertilization has been shown to alter local weed flora, favoring species better adapted to nutrient-rich environments [51].

Moreover, the integration of advanced technologies, such as IoT for monitoring agricultural fields, is optimizing weed identification and management, enabling more effective control strategies [52]. The use of drones and image recognition algorithms, as noted by Juwono et al. [48] and Kumar et al. [52], can facilitate real-time weed identification and classification, improving agricultural interventions and enhancing system sustainability.

### 4.2. Effects of Nitrogen Levels on Rice Plants

In terms of agronomic variables, nitrogen fertilization had the greatest impact on vegetative growth (VG), with an F-statistic of 33.079 and a highly significant *p*-value, confirming the strong effect of the applied treatments. These findings are consistent with those of Zhu et al. [50], who emphasized nitrogen’s critical role in promoting vegetative development, particularly in cereal crops, stimulating robust growth and greater biomass accumulation. This reinforces the importance of proper nitrogen management for maximizing nutrient use efficiency and improving crop yields [53].

The role of organic nitrogen sources must also be acknowledged. Recent research has shown that applying nitrogen fertilizers in combination with beneficial microorganisms, such as *Azoarcus* sp., promotes plant growth and enhances nutrient uptake efficiency, even at lower fertilization levels, while minimizing environmental impact [54]. However, prolonged fertilizer use, especially nitrogen and phosphorus, not only influences weed growth but also alters species composition. Studies indicate that combining NPK fertilization with organic matter reduces weed density without compromising species diversity, thus improving crop competitiveness and productivity [8].

Significant differences were also observed in variables such as the panicle number (PN) and days to flowering (DF), further highlighting the importance of fertilization management on crop reproductive development. These results emphasize the need for precise fertilizer applications to optimize nutrient use efficiency (NUE) and enhance crop performance [55,56]. Long-term fertilization in continuous cropping systems has been shown to reduce resource competition between crops and weeds by maintaining soil nutrient balance [16,46]. Integrating nutrient and weed management strategies is key to achieving long-term sustainability in intensive agricultural systems.

Partial correlation and principal component analysis revealed that the grains per panicle (GP) and moisture at harvest (MH) were the most central variables within the network, indicating their crucial role in interactions with other agronomic factors. The strong correlation between GP and MH suggests that a higher panicle density is associated with greater moisture retention at harvest, which could significantly impact grain quality and economic yield [55,57]. These results also align with studies on optimizing irrigation and fertilization practices, which influence root-associated bacterial community structures and enhance water and nutrient uptake, particularly in water-efficient systems [53,58].

PCA further demonstrated that variables such as the days to flowering (DF), plant height (PH), and yield per hectare (YHA) had the greatest influence on the first principal component (PC1), suggesting a trade-off between vegetative growth and yield. This finding is consistent with studies showing that high nitrogen fertilization levels promote vegetative growth at the expense of grain yield [45,59].

Nitrogen fertilization significantly impacts weed community dynamics and crop performance in rice systems. Research shows that high nitrogen availability intensifies competition between crops and weeds, favoring nutrient-demanding species such as *Echinochloa colona* and *Cyperus difformis*. This shift often reduces biodiversity, particularly in poorly managed systems reliant on nitrogen-heavy weed control strategies [60,61]. Balanced nitrogen applications, as evidenced in long-term studies, help mitigate these effects by promoting rice growth while suppressing weed biomass without eliminating weed species. Preserving diverse weed communities remains vital due to their contributions to soil structure and nutrient cycling [60,62]. These findings emphasize the importance of well-planned nitrogen management for achieving sustainable weed control in agricultural systems [62].

Beyond weed dynamics, nitrogen fertilization has long-term effects on soil health and sustainability. The excessive use of chemical nitrogen fertilizers is known to acidify soils and diminish microbial diversity, disrupting essential processes like nitrogen fixation and organic matter decomposition [61,63]. Integrated nutrient management, which combines organic and inorganic inputs, offers a viable solution by sustaining rice yields and enhancing soil organic carbon and microbial activity [9,63]. Such systems improve resilience to environmental stressors and promote ecological balance [64,65]. Transitioning to these approaches is essential for sustainable agriculture, ensuring productivity while safeguarding ecosystem services over the long term [9,61].

This work extends current agronomic research by illustrating how adjusting nitrogen levels can effectively manage both rice yield and weed incidence, tailored to Ecuador’s specific regional conditions. Beyond crop management, these findings underscore the broader environmental benefits of optimized nitrogen use, such as reduced herbicide reliance and improved nutrient efficiency. This perspective strengthens the contribution of our study, positioning it as a crucial tool for sustainable agricultural practices that balance productivity with ecological stewardship.

## 5. Conclusions

These research findings unequivocally demonstrate the substantial impact of nitrogen fertilization and planting distance on weed dynamics and rice crop performance in Ecuador. The predominance of Cyperaceae and Gramineae species within the weed communities across both study sites underscores the necessity for location-specific management strategies tailored to local agroecological conditions. In Guayas, sedges exhibited superior competitive ability, while grasses were more abundant in Los Ríos, reflecting the influence of soil and climatic variations on species distribution.

The implications of our findings are significant, contributing to theory by offering a refined model for nutrient–crop–weed interactions that could inform future agroecological studies. Practically, these results advocate for an integrative approach to nutrient management, balancing productivity gains with ecological considerations, which is essential for sustainable agriculture.

Future research could expand on these findings by addressing several critical gaps. Investigating the long-term effects of nitrogen fertilization on soil health, including changes in microbial diversity and soil structure, would provide a deeper understanding of its implications. Exploring alternative fertilization strategies, such as organic or biofertilizers, could offer insights into reducing environmental impacts while maintaining crop productivity. Additionally, integrating advanced agronomic variables, including precision agriculture tools, like digital sensors and remote monitoring, could enhance nutrient management and optimize rice cultivation. These advancements would not only support the development of more resilient and resource-efficient agricultural systems but also align with global efforts to promote sustainable farming practices.

## Figures and Tables

**Figure 1 life-14-01601-f001:**
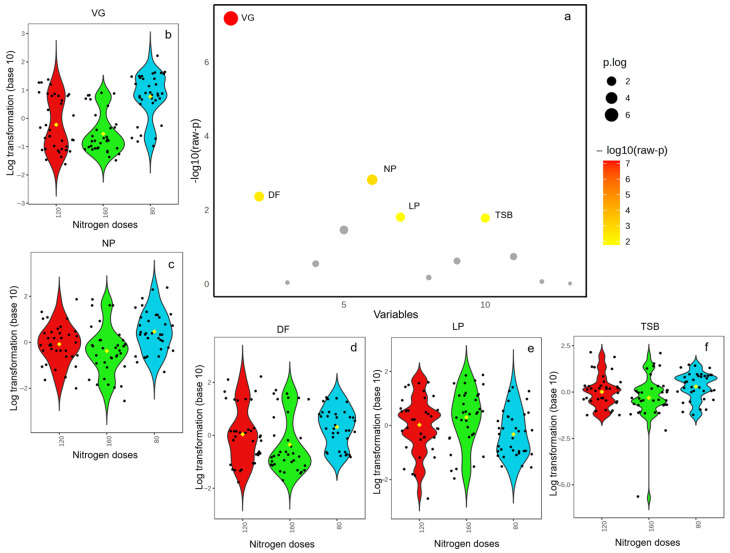
(**a**) Important variables selected by the Kruskal–Wallis test plot with a *p*-value threshold of 0.05. (**b**) Vegetative growth (days) (VG) across nitrogen fertilization levels; (**c**) NP: number of panicles (n); (**d**) DF: days to flowering (days); (**e**) LP: panicle length (cm); (**f**) TSB: 1000 seed weight.

**Figure 2 life-14-01601-f002:**
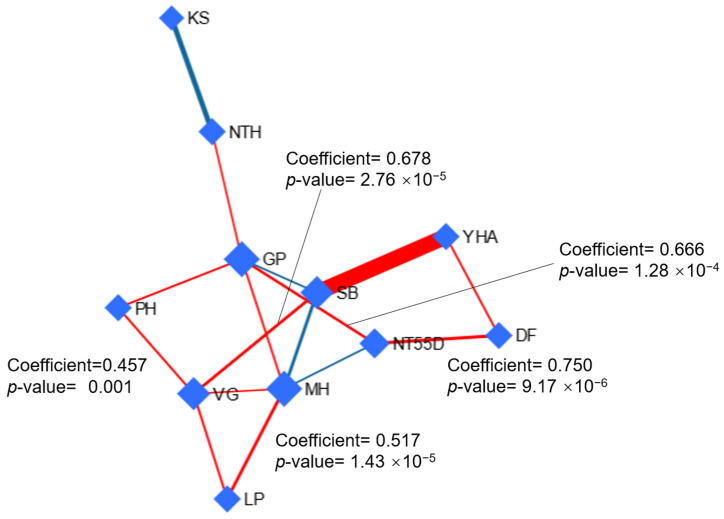
Debiased sparse partial correlation network. The *p*-value and correlation partial coefficient are shown for some important variables. Note: In the DSPC network, nodes are the input variables, while edges represent association measures. Variables with stronger associations are clustered together, and the edges between them are wider. Red lines show a positive correlation, while blue lines show a negative correlation with variables. VG: vegetative growth (days); DF: days to flowering (days); PH: plant height (cm); NT55D: number of tillers at 55 days (n); NTH: number of tillers at harvest (n); LP: panicle length (cm); GP: grains per panicle (n); KS: empty grains (percentage); MH: moisture at harvest (%); SB: sample biomass (g); YHA: yield per hectare (kg).

**Figure 3 life-14-01601-f003:**
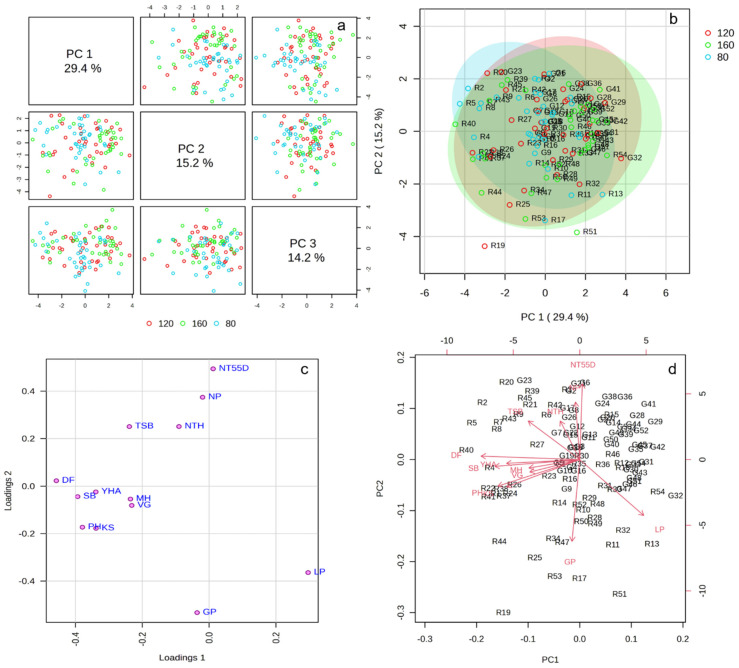
(**a**) Graph of pairwise scores between the two components. The explained variance of each component is shown in the corresponding diagonal cell; (**b**) score plot between the two PCs; (**c**) loading plot between the two PCs; (**d**) PCA biplot between the two components.

**Table 1 life-14-01601-t001:** Agronomic variables were measured to assess the vegetative growth, yield, and other characteristics of the rice plants.

Category	Variable	Abbreviation
Vegetative Growth	Vegetative cycle (days)	VG
	Days to flowering (days)	DF
	Plant height (cm)	PH
	Number of tillers at 55 days (n)	NT55D
	Number of tillers at harvest (n)	NTH
Yield Characteristics	Number of panicles (n)	NP
	Panicle length (cm)	LP
	Grains per panicle (n)	GP
	Empty grains (percentage)	KS
	1000 seed weight (g)	TSB
Other Attributes	Moisture at harvest (%)	MH
	Sample biomass (g)	SB
	Yield per hectare (kg)	YHA

**Table 2 life-14-01601-t002:** Weed incidence (%) (mean ± standard deviation) at the two locations.

	Weed Type (%)		
Location	Poaceae	Cyperaceae	Broadleaf
Virgen de Fatima, Guayas	39.22 ± 5.28 bB	46.27 ± 7.06 aA	14.61 ± 6.67 cA
Montalvo, Los Ríos	55.28 ± 5.56 aA	34.56 ± 5.61 bB	10.22 ± 5.28 cA

Note: Rows with different lowercase letters indicate statistical differences among the three weed groups at the same location (a, b, c). Columns with different uppercase letters indicate statistical differences in the two weed groups between the locations (A, B). Tukey’s test.

## Data Availability

The raw data supporting the conclusions of this article will be made available by the authors on request.

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
