# Peer review of "Fertilization for Growth or Feeding the Weeds? A Deep Dive into Nitrogen’s Role in Rice Dynamics in Ecuador"

_life, 2024, doi:10.3390/life14121601_

Round 1

Reviewer 1 Report

Comments and Suggestions for Authors

The manuscript entitled “Fertilizing for growth or feeding the weeds? A deep dive into nitrogen's role in rice dynamics in Ecuador” investigated the impact of nitrogen fertilization on rice growth and weed dynamics in Ecuador's main rice-producing regions. The paper started with an unclear abstract. The abstract section should be rewritten. The aim of the research is not clear, furthermore, the method is not sufficient. Some of the information related to data analysis (line 20) is not necessary to mention in the abstract section, otherwise, it should mention the abbreviation of the treatments. The result section is written in a general way, without any comparison between treatments and there is not any comparison with percentages. Please see more comments:

Line 40: what does the author mean for ..... source of employment in t, ..... ?????!!!!

I recommend adding some information about different rice farming methods like as conventional and organic rice farming on climate change and global warming. Here is a published work and you can use it for improve the cohesion of this section: https://doi.org/10.3390/su142315870

In paragraph four, add previous research on weed management and highlight the novelty of your research and explain what is the gap of knowledge in previous research and how your research can fill in this gap.

The hypothesis of the research is missing, please add it in the last paragraph of the introduction section.

Add the year of the field experiment in the material and method section.

There is no information about irrigation and also soil characteristics. Add these two in the material and method section.

Vertical axes are missing for Figures 2b,c,d,e.

It’s better to add the abbreviation for the Figures (1a-e) in the Figure caption. (mention abbreviations in Table 1 is urgent but re mention in Figures captions help readers for more understanding_.

Add legend for Figure 2. Also, add abbreviations in the caption, because it helps readers for more understanding.

I recommend dividing the discussion section like results into some subsections and discussing each section separately.

The conclusion is well rewritten.

Author Response

Comments 1: The paper started with an unclear abstract. The aim of the research is not clear. Some of the information related to data analysis (line 20) is not necessary to mention in the abstract section it should mention the abbreviation of the treatments.

Responses 1: The observations made by Reviewer 1 were addressed, and changes were made to the abstract to enhance its clarity and understanding.

Rice (Oryza sativa L.) is a crucial crop for employment and agricultural output, heavily reliant on family labor. This study evaluated the effects of nitrogen levels (80, 120, and 160 kg·ha⁻¹) on weed incidence and key agronomic variables, including vegetative growth, yield, and related traits, in Ecuador's primary rice-growing regions, Guayas and Los Ríos. A split-plot randomized complete block design was implemented using two rice varieties (INIAP-FL-Elite and SFL-11) and three planting densities (20x30, 25x30, and 30x30 cm). Weed incidence was higher in Los Ríos, dominated by grasses (55.28%), while Cyperaceae (46.27%) prevailed in Guayas. Data analysis included non-parametric tests to identify significant treatment effects, debiased sparse partial correlations (DSPC) to reveal key agronomic interactions, and Principal Component Analysis (PCA) to identify influential traits, ensuring robust and normalized interpretations. Analysis of variance indicated significant effects for all variables, with vegetative growth (VG) most affected (P<0.001). The number of panicles (NP) and days to flowering (DF) showed significant, though less pronounced, effects, while panicle length (LP) and 1000-seed biomass (TSB) exhibited moderate responses. DSPC highlighted grains per panicle (GP) and total biomass (SB) as critical variables, with significant correlations between days to flowering and tiller count at 55 days (r= 0.750, P<0.001) and between vegetative growth and total biomass (r= 0.678, P<0.001). PCA explained 58.8% of the total variance, emphasizing days to flowering, plant height, total biomass, and yield as the most influential traits. These findings underline the importance of integrated nutrient and weed management strategies tailored to Ecuador's agroecological conditions.

Comments 2: what does the author mean for ..... source of employment in t, ..... ?????!!!!

Responses 1: Regarding this abbreviation: t = tons.

In Ecuador, there are considerable rice germplasm, comprising publicly released rice varieties, nursery observation lines and landraces varieties [1]. Actually 317,400 ha of rice (Oriza sativa L.) are cultivated, and distributed across the provinces of Guayas, Los Ríos, Manabí, Loja, and El Oro, with a total production of 1,697,353 tons, where Los Ríos and Guayas provinces provide 95% of the national output [2].

Comments 3: I recommend adding some information about different rice farming methods like as conventional and organic rice farming on climate change and global warming. https://doi.org/10.3390/su142315870

Responses 3: In response to this comment, two paragraphs were added, based on the review of the suggested article.

Conventional and organic rice farming systems differ significantly in their environmental impacts, particularly in the context of climate change. Conventional agriculture, heavily dependent on chemical inputs such as synthetic fertilizers and pesticides, is a major contributor to greenhouse gas (GHG) emissions, including carbon dioxide (CO₂), methane (CH₄), and nitrous oxide (N₂O); these emissions are notably exacerbated in flood-irrigated systems, which promote CH₄ production ([4,5]. The same authors mentioned that, in contrast, organic farming practices reduce CO₂ emissions by approximately 32.7% compared to conventional methods through the use of natural fertilizers, such as vermicompost and livestock manure. However, during the initial transition to organic systems, higher emissions of N₂O and CH₄ may occur due to the decomposition of organic matter ([4,5].

From a sustainability perspective, organic practices offer several environmental benefits. They lower environmental toxicity, mitigate freshwater eutrophication, reduce fossil fuel dependency, and improve soil properties and microbial diversity. These practices also enhance nutrient use efficiency while minimizing emissions of ammonia (NH₃), sulfur dioxide (SO₂), and nitrogen dioxide (NO₂) [6]. Although organic systems require time to achieve stabilization, they represent a viable alternative for mitigating climate change and fostering a resilient global agricultural framework ([5].

Comments 4:  add previous research on weed management and highlight the novelty of your research and explain what is the gap of knowledge in previous research and how your research can fill in this gap.

Responses 4: In response to this comment, one paragraph was added, based on the review of the suggested.

In Ecuador in rice field áreas close of the Guayas river basin, used commonly herbicides such as butachlor and pendimethalin to control grassy weeds (Deknock et al., 2019) [20], that are applied pre-emergence or post-emergence in flooded rice fields or direct seeded. According to Nazir et al. [21] several systems in rice are used, as transplanting rice, direct seeded and system of intensification, where penoxsulam herbicide recorded highest grain yield and was effective in reducing weed populations and its dry biomass per meter square.

It is hypothesized that nitrogen fertilization in rice fields enhances crop growth and, when combined with appropriate planting distances and effective weed management strategies, can mitigate the competitive impact of weeds on yield by reducing their incidence.

Comments 6: The method is not sufficient. Add the year of the field experiment in the material and method section. There is no information about irrigation and also soil characteristics.

Responses 6: The suggestions from reviewer regarding the materials and methods section were accepted.

The research was conducted in two rice-growing areas of Ecuador during the dry season of 2022 (August to December). The locations were: Virgen de Fátima Parish, Yaguachi Municipality, Guayas Province, located in the South Coast Experimental Station (INIAP), between the coordinates 2°15'15" S and 79°30'40" W and in the Montalvo Parish, Babahoyo Municipality, Los Ríos Province, located in the CEDEGE area, at coordinates 1°53'31.38" S and 79°27'27.31" W.

The factors evaluated were localities (Virgen de Fatina and Montalvo), nitrogen doses at three levels (80, 120, and 160 kg·ha⁻¹), rice varieties (O. sativa) at two levels (INIAP-FL-Elite and SFL-11), and planting distances at three levels (20x30, 25x30, and 30x30 cm). These combinations of nitrogen doses, rice varieties, and planting distances formed the treatment groups.

Prior to the application of treatments, weeds in the study plots were classified based on leaf width up to the family level, as reproductive structures were absent at the time of evaluation. Narrow-leaved weeds were identified and categorized under the botanical families Cyperaceae and Gramineae. Furthermore, broad-leaved weeds were grouped and classified based on their structural characteristics as herbaceous, semi-woody, or woody.

Before land preparation, soil sampling was performed at both the experimental site for physical and chemical analysis. A zigzag sampling method was used, with samples taken from several points in the field to create a composite sample, ensuring representativeness. Analyses were conducted at the National Institute for Agricultural Research, El Boliche Experimental Station. In the both sites the soil is clay loam.

To establish the seedbeds, 3 kg of certified seed was used for each variety under study. Seedbeds were prepared with an area of 10 m², and 300 g of seed per m² were distributed in each bed. Vydate (2 cc·m-2) was applied to prevent nematode damage. Fertilization was performed on the seedbeds 12 days after sowing (DAS) with urea applied at a rate of 20 g·m-2. Before transplanting, the seedbeds were exposed to a water layer of approximately 5 to 8 cm for 4 days to facilitate seedling removal and reduce root damage. Transplanting was carried out when the plants were 25 days old in flooded soil without water lamina to prevent damage from the apple snail (Pomacea canaliculata). Between 3 and 4 plants per planting site were transplanted, considering the planting distances of this study (20x30, 25x30, and 30x30 cm).

Comments 7: without any comparison between treatments and there is not any comparison with percentages

Responses 7: Although the reviewer indicates that percentages are not compared, the results reflect the percentage changes in the two studied locations. Additionally, some clarifications were made regarding the two locations.

In both localities, three main categories of weeds were observed: grasses, sedges, and broadleaves. However, notable differences in their incidence were detected between the two areas. Statistically significant differences (P<0.001) were found among these groups in Virgen de Fatima, Guayas, where sedges constituted 46.27% of the species, grasses 39.22%, and broadleaves 14.61%. In contrast, in Montalvo, Los Ríos, grasses dominated (55.28%), followed by sedges (34.56%) and broadleaves (10.22%), with similarly significant statistical differences (P<0.001) among the three groups (Table 2).

Comments 8: It’s better to add the abbreviation for the Figures (1a-e) in the Figure caption.

Responses 8: The suggested changes by Reviewer 1 were made to the figures.

Figure 1. a) Important variables selected by Kruskal-Wallis Test plot with p-value threshold 0.05; b) Vegetative growth (days) (VC) across nitrogen fertilization levels; c) NP: Number of panicles (n); d) DF: Days to flowering (days); e) LP: Panicle length (cm); f) TSB: 1000 seed weight.

Furthermore, the number of tillers at 55 days (NT55D) had a strong positive relationship with grains per panicle (GP) (0.666, P= 1.28E-04), underscoring the role of early tiller development in determining grain yield potential. The significant p-values associated with these partial correlations (P<0.05) provide robust statistical support for these interactions, reinforcing the importance of GP, MH, and VC as central variables in the network. These findings suggest that these key agronomic traits are likely influenced by nitrogen fertilization levels and could serve as critical targets for optimizing rice growth and yield in future studies (Figure 2).

Figure 2. Debiased sparse partial correlation network. The P-value, and correlation partial coefficient are shown for some important variables. Note: In the DSPC network, nodes are the input variables, while edges represent association measures. Variables with stronger associations are clustered together and the edges between them are wider. Red lines show a positive correlation, while blue lines show a negative correlation with variables. VC: Vegetative growth (days), DF: Days to flowering (days), PH: Plant height (cm), NT55D: Number of tillers at 55 days (n), NTH: Number of tillers at harvest (n), LP: Panicle length (cm), GP: Grains per panicle (n), KS: Empty grains (percentage), MH: Moisture at harvest (%), SB: Sample biomass (g), YHA: Yield per hectare (kg).

Comments 9: I recommend dividing the discussion section like results into some subsections and discussing each section separately

Responses 9: The reviewer’s suggestion was implemented, and the discussion was reorganized to align with the results, with each one discussed separately.

4.1. Levels of Weed Incidence

4.2. Effect of nitrogen levels on rice plants

Comments 10: The conclusion is well rewritten.

Responses 10: We appreciate Reviewer 1's comments on the conclusions; however, a new paragraph was added to incorporate Reviewer 2's suggestions.

Reviewer 2 Report

Comments and Suggestions for Authors

Title:

Fertilizing for growth or feeding the weeds? A deep dive into nitrogen's role in rice dynamics in Ecuador

Recommendation:

    Minor revision.

Comments:

This manuscript investigates the effects of nitrogen fertilization on rice cultivation in Ecuador, focusing on its impact on weed dynamics and rice productivity. It highlights the challenges posed by increased nitrogen use, which can inadvertently promote weed growth, leading to reduced crop yields and higher herbicide costs. The study aims to fill gaps in existing research by examining how different levels of nitrogen fertilization influence specific weed species and rice performance in Ecuador's unique agroecosystems. Advanced analytical methods, such as Debiased Sparse Partial Correlations (DSPC) and the Kruskal-Wallis test, are employed to analyze the complex interactions between nitrogen levels, weed competition, and rice growth, ultimately seeking to develop more sustainable agricultural practices. The subject is relevant and consistent with the aims and scopes of the journal. Several comments and suggestions are offered below with the intent to assist the author in improving the manuscript.

1.      In the manuscript, the section "2. Materials and Methods" should be reorganized with numbered subsections (e.g., 2.1, 2.2, etc.) and rewritten accordingly for clarity and structure.

2.      The format and content of the references contain numerous errors and require careful correction.

3.      In this manuscript, while the use of advanced analytical methods such as Debiased Sparse Partial Correlations (DSPC) and the Kruskal-Wallis test is commendable, the article should provide more detailed information on the experimental design, including sample sizes, replication, and control measures. This would enhance the reproducibility of the study and allow for a better assessment of the results' validity.

4.      In the manuscript, the conclusion could be expanded to include suggestions for future research. Identifying gaps in the current study and proposing areas for further investigation, such as the long-term effects of nitrogen fertilization on soil health or the exploration of alternative fertilization strategies, would provide a comprehensive outlook and encourage ongoing research in this field.

5.      The study primarily addresses immediate effects of nitrogen fertilization on rice and weeds. However, it could be improved by discussing the long-term implications of these practices on soil health, biodiversity, and sustainability. This would provide a more balanced view of the potential consequences of intensive nitrogen use.

Author Response

The article should provide more detailed information on the experimental design, including sample sizes, replication, and control measures.

Responses 1: The suggestions from both reviewers regarding the materials and methods section were accepted, and the activities conducted have been separated and detailed.

2.1. Location and Study Period

The research was conducted in two rice-growing areas of Ecuador during the dry season of 2022 (August to December). The locations were: Virgen de Fátima Parish, Yaguachi Municipality, Guayas Province, located in the South Coast Experimental Station (INIAP), between the coordinates 2°15'15" S and 79°30'40" W and in the Montalvo Parish, Babahoyo Municipality, Los Ríos Province, located in the CEDEGE area, at coordinates 1°53'31.38" S and 79°27'27.31" W.

2.2. Treatments in study

The factors evaluated were localities (Virgen de Fatina and Montalvo), nitrogen doses at three levels (80, 120, and 160 kg·ha⁻¹), rice varieties (O. sativa) at two levels (INIAP-FL-Elite and SFL-11), and planting distances at three levels (20x30, 25x30, and 30x30 cm). These combinations of nitrogen doses, rice varieties, and planting distances formed the treatment groups.

Prior to the application of treatments, weeds in the study plots were classified based on leaf width up to the family level, as reproductive structures were absent at the time of evaluation. Narrow-leaved weeds were identified and categorized under the botanical families Cyperaceae and Gramineae. Furthermore, broad-leaved weeds were grouped and classified based on their structural characteristics as herbaceous, semi-woody, or woody.

2.3. Experimental Design

One hundred eight plots were used for the study in each locality, each plot covering a total area of 9.00 m². The width of the plots was 1.80 m, and the length was 5.00 m. The usable area for each plot was 6.00 m², with a separation of 1.5 m between blocks, corresponding to a total experimental area of 162 m².

A randomized complete block design with split-plot treatments was employed, with the main plot representing the location and a full factorial arrangement (2 x 3 x 2 x 3) applied in the subplots with three replications. The main plot was composed of the location (two levels), while the factors and levels within the subplots included nitrogen, rice varieties, and planting distances as previously mentioned.

2.4. Management of the Trial

Before land preparation, soil sampling was performed at both the experimental site for physical and chemical analysis. A zigzag sampling method was used, with samples taken from several points in the field to create a composite sample, ensuring representativeness. Analyses were conducted at the National Institute for Agricultural Research, El Boliche Experimental Station. In the both sites the soil is clay loam.

To establish the seedbeds, 3 kg of certified seed was used for each variety under study. Seedbeds were prepared with an area of 10 m², and 300 g of seed per m² were distributed in each bed. Vydate (2 cc·m-2) was applied to prevent nematode damage. Fertilization was performed on the seedbeds 12 days after sowing (DAS) with urea applied at a rate of 20 g·m-2. Before transplanting, the seedbeds were exposed to a water layer of approximately 5 to 8 cm for 4 days to facilitate seedling removal and reduce root damage. Transplanting was carried out when the plants were 25 days old in flooded soil without water lamina to prevent damage from the apple snail (Pomacea canaliculata). Between 3 and 4 plants per planting site were transplanted, considering the planting distances of this study (20x30, 25x30, and 30x30 cm).

Weed control was performed 10 days after transplanting (DAT) when weeds had developed 2 to 3 leaves. A chemical control method was employed, using selective herbicides for rice cultivation, applying Butachlor + Pendimethalin + Bispyribac at rates of 40 + 25 + 01 L commercial product·ha⁻¹, respectively. Then, weed monitoring was conducted at 30, 45, and 60 days after herbicide application (DAA). Fertilization was applied at 15, 30, and 45 days after transplanting (DAT), with nitrogen sources provided according to the nitrogen levels specified in the treatments. Harvesting occurred 120 days into the crop cycle when the materials reached physiological maturity.

2.5. Evaluation of Study Variables

Thirteen agronomic variables were evaluated to examine the vegetative growth, yield performance, and other characteristics of the rice plants (Table 1). The vegetative cycle was expressed in days (VC), considering the time lapse from seedling emergence to the harvest of all experimental units. The variables evaluated included the number of days to flowering (DF), counted from transplanting to the appearance of the first panicles. Plant height (PH) was measured before harvest in 10 randomly selected plants from the usable area of each experimental unit, measured from the plant collar to the apex of the most prominent panicle, with data expressed in cm. The number of tillers was counted 55 days after transplanting (NT55D) and at harvest (NHT) in the 10 selected plants. The number of panicles (NP) was recorded, and panicle length (LP) was measured from the ciliary node to the panicle apex, excluding awns. Measurements were made on panicles from the 10 selected plants. Grain sterility percentage was determined on the panicles selected for the panicle length variable, and the total number of fertile (filled) (GP) and sterile (empty) grains (KS) was counted. Results were expressed as percentages. Finally, 1,000 grains from each treatment were counted and weighed (TSB) using an analytical balance, with the results expressed in grams.

Table 1. Agronomic variables were measured to assess the vegetative growth, yield, and other characteristics of rice plants.

Category

Variable

Abbreviation

Vegetative Growth

Vegetative cycle (days)

VC

Days to flowering (days)

DF

Plant height (cm)

PH

Number of tillers at 55 days (n)

NT55D

Number of tillers at harvest (n)

NTH

Yield Characteristics

Number of panicles (n)

NP

Panicle length (cm)

LP

Grains per panicle (n)

GP

Empty grains (percentage)

KS

1000 seed weight (g)

TSB

Other Attributes

Moisture at harvest (%)

MH

Sample biomass (g)

SB

Yield per hectare (kg)

YHA

Each plot was harvested when harvest the plants complete a cycle productive and was measured weight (WS) and moisture content (MH) of the sample. Subsequently, the yield (YHA) was expressed in kilograms per hectare, estimated although the formula WAT= [(WS*(100 – IM)/(100 – DM)) * (10/HA)]; where: WS= weight of the sample (g), IM= initial moisture content; DM= desired moisture content of 14%; HA= harvested area (m2); WAT= weight adjusted to the treatment.

2.6. Data Analysis

Comments 2: The study primarily addresses immediate effects of nitrogen fertilization on rice and weeds. However, it could be improved by discussing the long-term implications of these practices on soil health, biodiversity, and sustainability. This would provide a more balanced view of the potential consequences of intensive nitrogen use.

Responses 2: The observations made by reviewer were addressed, the discussion was reorganized according to the results presented, and a significant portion of their suggestions were incorporated.

4.1. Levels of Weed Incidence

4.2. Effect of nitrogen levels on rice plants

Nitrogen fertilization significantly impacts weed community dynamics and crop performance in rice systems. Research shows that high nitrogen availability intensifies competition between crops and weeds, favoring nutrient-demanding species such as Echinochloa colona and Cyperus difformis. This shift often reduces biodiversity, particularly in poorly managed systems reliant on nitrogen-heavy weed control strategies ​[60,61). Balanced nitrogen applications, as evidenced in long-term studies, help mitigate these effects by promoting rice growth while suppressing weed biomass without entirely eliminating weed species. Preserving diverse weed communities remains vital due to their contributions to soil structure and nutrient cycling​ [60,62]. These findings emphasize the importance of well-planned nitrogen management for achieving sustainable weed control in agricultural systems​ [62].

Beyond weed dynamics, nitrogen fertilization has long-term effects on soil health and sustainability. Excessive use of chemical nitrogen fertilizers is known to acidify soils and diminish microbial diversity, disrupting essential processes like nitrogen fixation and organic matter decomposition​ [61,63]. Integrated nutrient management, which combines organic and inorganic inputs, offers a viable solution by sustaining rice yields and enhancing soil organic carbon and microbial activity [9,63]. Such systems improve resilience to environmental stressors and promote ecological balance. Transitioning to these approaches is essential for sustainable agriculture, ensuring productivity while safeguarding ecosystem services over the long term [9,61].

Comments 3: In the manuscript, the conclusion could be expanded to include suggestions for future research. Identifying gaps in the current study and proposing areas for further investigation, such as the long-term effects of nitrogen fertilization on soil health or the exploration of alternative fertilization strategies, would provide a comprehensive outlook and encourage ongoing research in this field.

Responses 3: a new paragraph was added to incorporate suggestions.

Future research could expand on these findings by addressing several critical gaps. Investigating the long-term effects of nitrogen fertilization on soil health, including changes in microbial diversity and soil structure, would provide a deeper understanding of its implications. Exploring alternative fertilization strategies, such as organic or biofertilizers, could offer insights into reducing environmental impacts while maintaining crop productivity. Additionally, integrating advanced agronomic variables, including precision agriculture tools like digital sensors and remote monitoring, could enhance nutrient management and optimize rice cultivation. These advancements would not only support the development of more resilient and resource-efficient agricultural systems but also align with global efforts to promote sustainable farming practices.

Lastly, in the manuscript text, the included observations are highlighted in a different color, and we are open to any further comments the reviewers may deem pertinent.

Round 2

Reviewer 1 Report

Comments and Suggestions for Authors

.